# Spectrum-guided Multi-view Graph Fusion

## Abstract

Multi-view graphs capture diverse relations among entities through graph views and individual characteristics via attribute views, presenting a challenge for unsupervised learning due to potential conflicts across views. Existing approaches often lack efficacy, efficiency, and the ability to explicitly control view contributions. In this paper, we present SMGF, a novel graph fusion framework that approximates underlying entity connections by aggregating view-specific graph structures. We construct a multi-view Laplacian $\mathcal{L}$ from normalized Laplacian matrices representing all views. View weights are determined through the optimization of two objectives derived from $\mathcal{L}$'s spectral properties, which exploit the eigenvalue gap and enhance connectivity. Comprehensive experiments on six real-world datasets showcase the superior performance of SMGF in node embedding and clustering results, along with its efficiency and scalability. SMGF offers a promising solution for unsupervised learning on multi-view graphs, addressing the challenge of interpretably combining diverse and potentially conflicting information from both graph and attribute views. The source code of SMGF is available at `https://anonymous.4open.science/r/SMGF-E903/`.

## 1 Introduction

Real-world entities can be characterized from multiple viewpoints. For instance, in a complex social network, diverse interpersonal connections, including friendships, familial ties, and professional affiliations, are modeled by separate graph views. Each individual could also be described by a wide range of attribute views, such as demographic statistics, facial appearance features, and behavioral characteristics. Multi-view graph data are the combination of these graph views and attribute views. Such datasets have proven highly useful for recommendation systems (Wang et al., 2020), image processing (Nie et al., 2018), and bioinformatics (Fu et al., 2021), among others.

In this work, our primary focus is unsupervised learning over multi-view graph data, where both graph views and attribute views are present. Specifically, we aim for node representation learning and clustering tasks. Despite abundant research on unsupervised graph learning, it remains a challenging problem for multi-view graph data. Though multiple views offer rich insights from distinct perspectives, their inherent diversity inevitably results in inconsistency. An unsupervised algorithm must attain a consensus based on potentially contradictory information, all without prior knowledge of the relative importance associated with these views. Moreover, some views may contain noisy data, and graph views are often incomplete.

A plethora of research has been conducted on the multi-view clustering (MVC) problem (Fang et al., 2023), where the input consists of purely attribute views that are extracted from webpages, visual features, etc. In recent years, graph views from real-world networks have also been incorporated into unsupervised learning on multi-view graph data. A few existing works leverage graph neural network (GNN) models such as graph autoencoder (Fan et al., 2020) and deep graph infomax (Park et al., 2020). These deep learning approaches suffer from a lack of interpretability and reduced efficiency caused by the large number of model parameters they entail. Additionally, their capabilities are constrained to handle only a single view of node attributes. Another line of work performs graph filtering on attributes and subsequently finds a consensus graph (Pan & Kang, 2021) or low-dimensional representation (Lin & Kang, 2021). Their optimization processes usually rely on the assumption that all graph views exhibit some degree of adherence to a shared cluster structure, a presumption that can be overly stringent for real-world datasets. Empirically, we note that achiev-

ing optimal clustering performance with these methods requires meticulous hyperparameter tuning efforts across different datasets.

In this paper, we present SMGF, a novel framework for unsupervised learning over multi-view graphs. Essentially, SMGF framework adopts a graph fusion mechanism and constructs a multi-view Laplacian $\mathcal{L}$ with desired spectral properties via weighted aggregation of single-view Laplacian matrices. In the initial stage, all views undergo projection to normalized graph Laplacians. This unification step effectively merges both graph and attribute views into a common domain for graph fusion. To this end, it is assumed the true underlying graph structure approximates a weighted linear aggregation of these single-view Laplacians. In the following stage, the suitable view weights are determined by a spectrum-guided optimization scheme with two objectives. The eigengap objective exploits the dataset's inherent class count, while the connectivity objective addresses the challenges posed by incompleteness and irregularity in graph views. Both objectives focus only on the spectral properties of the fused multi-view Laplacian $\mathcal{L}$, thereby distinguishing our approach from prior works that rely on assumptions concerning individual views. In addition, the resulting view weights provide a clear indication of the individual contribution of each view, thus augmenting the interpretability of the obtained results.

After constructing the multi-view Laplacian $\mathcal{L}$ that represents graph fusion, we can directly perform spectral clustering or attain node embeddings via matrix factorization. We demonstrate the unsupervised learning capabilities of our proposed approach (SMGF) through comprehensive experimentation on real-world datasets. Node embeddings obtained through factorization of $\mathcal{L}$ consistently outperform alternative approaches in quality and efficiency, as evidenced by the evaluation results of the node classification task. By applying spectral clustering to $\mathcal{L}$, SMGF exhibits remarkable clustering quality compared to baseline methods. Notably, our results on the million-scale MAG dataset underscore the outstanding scalability of our approach. We also perform an exhaustive analysis on SMGF regarding the effect of alternative optimization objectives and various hyperparameter configurations.

We summarize the contributions of this work as follows:

- We study the problem of unsupervised learning on multi-view graphs and present SMGF, as a novel approach that addresses performance and interpretability with a graph fusion framework.

- From the spectrum of multi-view Laplacian, we formulate eigengap and connectivity objectives to characterize a desirable graph fusion. An optimization scheme is then designed to determine suitable view weights.

- Through extensive node embedding and clustering experiments, we demonstrate the superior unsupervised learning capabilities of our graph fusion framework.

## 2 PRELIMINARIES

A multi-view graph $\mathcal{G} = \{G_1, \ldots, G_a, X_1, \ldots, X_b\}$ consists of $a$ graph views $\{G_1, \ldots, G_a\}$ and $b$ attribute views $\{X_1, \ldots, X_b\}$. All $z = a + b$ views in $\mathcal{G}$ share the same node set $V = \{v_1, \ldots, v_n\}$. We focus on multi-view graphs with both graph and attribute views, *i.e.*, $a \geq 1$, $b \geq 1$, and $z \geq 3$.

Each graph view $G \in \{G_1, \ldots, G_a\}$ is an undirected graph without self-loops. A graph view $G = \{V, E\}$ with $n$ nodes and $m$ weighted edges can be represented using an adjacency matrix $A_G \in \mathbb{R}^{n \times n}$ which comprises $2m$ nonzero entries. Matrix entry $A_G[i, j]$ is the weight of edge $(v_i, v_j)$. The degree of node $v_i$ is defined as $\sum_{j=1}^{n} A_G[i, j]$. Attribute view $X \in \{X_1, \ldots, X_b\}$ is an $n \times d_X$ matrix where each row vector $x_i$ contains attribute values associated with node $v_i$.

For multi-view graphs, node embedding aims to learn a function $V \to \mathbb{R}^h$ that maps each node $v_i \in V$ to a latent representation vector. Despite the low dimensionality $h$, node embeddings must effectively capture the structural characteristics inherent to graph views and encode the information contained within attribute views. We evaluate the quality of embeddings on node classification task.

Given the number of clusters $k$, clustering over multi-view graph $\mathcal{G}$ aims to divide the $n$ nodes in $V$ into $k$ disjoint non-empty subsets $\{C_1, \ldots, C_k\}$, such that nodes within each cluster are densely connected in the graph views and share similar attributes in the attribute views.

## 3 METHODOLOGY

In this section, we describe `SMGF`, our novel framework for unsupervised learning over multi-view graphs. Our approach unfolds in three stages. In the initial stage, we unify all views by transforming them into Laplacian matrices, bringing them into a common normalized space. Subsequently, the second stage formulates the multi-view graph fusion as an optimization problem guided by eigenvalue-based objectives. Finally, we cover the implementation of node embedding and clustering, which leverage the multi-view Laplacian to derive node representations and clustering results.

### 3.1 PROJECTING VIEWS TO LAPLACIANS

Previous approaches to multi-view graph learning often rely on graph propagation techniques, such as graph filtering (Pan & Kang, 2021) or GNN models Fan et al. (2020). Nevertheless, questions about the contribution of each view to learned representations and the ability to adjust the importance of a specific view remain unanswered. We attribute this lack of interpretability and flexibility to the divergent treatment of graph views and attribute views. To address these issues, we propose a novel approach: projecting all views into a single normalized space of graph representations. This transformation allows for explicit weighting of views, regardless of their original data type.

$K$-nearest neighbor (KNN) graphs can effectively model the local neighborhood of data points, with applications in unsupervised learning problems such as spectral clustering (von Luxburg, 2007) and attributed network clustering (Li et al., 2023). Thus, we can construct a KNN graph for each attribute view and encode by graph Laplacian. In summary, each view in a multi-view graph is encoded into graph Laplacian as follows.

- Graph view $G$ with adjacency matrix $A_G$: Denote its diagonal node degree matrix by $D_V = diag(A_G \mathbf{1}_n)$. The normalized Laplacian is given by

$$L(G) = I - D_V^{-\frac{1}{2}} A_G D_V^{-\frac{1}{2}}. \tag{1}$$

- Attribute view $X$: For each attribute view $X$, we create a corresponding KNN graph where each node $v_i$ represents the attribute vector $x_i$ in $x$. An undirected graph $G_X$ is obtained by adding the KNN graph's adjacency matrix by the transpose. The attribute view is thus encoded by the normalized Laplacian $L(G_X)$.

**Graph fusion.** Given a multi-view graph $\mathcal{G}$ with $z = a + b$ views, we represent the $i$-th view using the normalized Laplacian matrix $L_i$. Since each view provides insight into the relationships among entities from a specific perspective, we hypothesize that the true underlying relationships among entities can be considered as a certain combination of these individual views. Consequently, we use a weighted graph fusion mechanism that directly aggregates the single-view Laplacians as follows.

$$\mathcal{L} = \sum_{i=1}^{z} w_i L_i, \text{ where } \sum_{i=1}^{z} w_i = 1. \tag{2}$$

Since the normalized Laplacian matrix of any graph is symmetric positive semi-definite, it follows that matrix $\mathcal{L}$ preserves this property, and thus its eigenvalues are nonnegative. Sorted in ascending order, the eigenvalues of $\mathcal{L}$ are $0 \leq \lambda_1 \leq \lambda_2 \leq \cdots \leq \lambda_n$. In this work, we refer to $\mathcal{L}$ as the multi-view Laplacian, despite the absence of a guaranteed graph Laplacian property $\lambda_1 = 0$. Nonetheless, we treat $\mathcal{L}$ as a pseudo graph Laplacian, *i.e.*, an approximation of $L(G_F)$. Here, $G_F$ signifies the underlying graph that encompasses all views contributing to the formation of $\mathcal{L}$.

### 3.2 SPECTRUM-GUIDED VIEW WEIGHTING

Each $w_i$ in Eq. (2) quantifies the contribution of the $i$-th view to the graph fusion. `SMGF` determines view weights by optimizing two objectives derived from eigenvalues of multi-view Laplacian $\mathcal{L}$.

#### 3.2.1 EIGENGAP OBJECTIVE

Given that the entities represented by the multi-view graph $\mathcal{G}$ are categorized into $k$ classes, we assume the nodes in $\mathcal{G}$ could be organized into a unified network $G_F$ with $k$ cohesive clusters. Consider a "perfectly-clustered" graph composed of $k$ disjoint complete subgraphs as an extreme

case. Its block-diagonal normalized Laplacian matrix has zero-valued eigenvalues of multiplicity $k$. The eigengap $\lambda_{k+1}/\lambda_k$ is infinitely large. For general graphs, a substantial eigengap between successive eigenvalues is a heuristic for determining the number of clusters (von Luxburg, 2007; Afzalan & Jazizadeh, 2019). In spectral graph theory, Lee et al. (2014) establish a formal connection between the eigengap and cluster quality by demonstrating an upper bound for the normalized cut.

**Definition 1** *For a cluster $C \subset V$ within graph $G$, its volume is $Vol(C) = \sum_{v_i \in C_i} d(v_i)$. $Cut(C) = \sum_{v_i \in C, v_j \notin C} A_G[i,j]$ is the total weight of outgoing edges from nodes within $C$. The normalized cut of $C$ is defined as $NCut(C) = Cut(C)/Vol(C)$.*

**Theorem 1** *(Lee et al., 2014) There is a constant $c > 0$ such that for any weighted graph $G$ and $k \in \mathbb{N}$, the following holds. Let $\delta \in (0, \frac{1}{3})$ be such that $\delta k$ is an integer. If $\lambda_{(1+\delta)k} > c\frac{(\log k)^2}{\delta^9}\lambda_k$, there are at least $r \geq (1 - 3\delta)k$ nonempty disjoint sets of nodes $C_1, C_2, \ldots, C_r \subseteq V$ such that $NCut(C_i) \lesssim \sqrt{\frac{\lambda_k}{\delta^3}}$.*

Let $\delta = 1/k$, and it follows from Theorem 1 that an asymptotic upper bound exists for the NCut of $r \geq k - 3$ disjoint clusters if an eigengap $\lambda_{k+1}/\lambda_k > ck^9(\log k)^2$ is present.

Considering the presence of $k$ ground truth classes within the multi-view graph, we assume a significant eigengap $\lambda_{k+1}/\lambda_k$ exists in the underlying graph $G_F$. To align the multi-view Laplacian $\mathcal{L}$ with the true class distribution, we propose maximizing the eigengap objective $f_{GAP} = \lambda_{k+1}(\mathcal{L})/\lambda_k(\mathcal{L})$ over valid weight variables subject to constraints in Eq. (5).

$$\max_w \lambda_{k+1}(\sum_{i=1}^{z} w_i L_i)/\lambda_k(\sum_{i=1}^{z} w_i L_i) \tag{3}$$

### 3.2.2 CONNECTIVITY OBJECTIVE

In real-world multi-view graph $\mathcal{G}$, graph views are often incomplete, where connections are missing for certain nodes. For instance, in the ACM dataset's co-author view, there are 156 connected components and 561 unconnected nodes out of a total of 3025 nodes. If an incomplete view is assigned a predominant weight, it could lead to a situation where the resulting $\mathcal{L}$ exhibits a large eigengap, despite the limited information it captures.

To mitigate this issue, we propose to promote the level of connectivity in the graph fusion $G_F$. Graph conductance $\Phi(G)$ is a common metric for graph connectivity, defined as the minimum $NCut(C)$ of any node set $C \subset V$ such that $Vol(C) \leq Vol(V)/2$. In spectral graph theory, Cheeger's inequality (Alon & Milman, 1985) bounds conductance with the second smallest eigenvalue $\lambda_2$ of $L(G)$.

**Theorem 2** *(Spielman, 2007) Let $\lambda_2$ be the second smallest eigenvalue of the normalized Laplacian matrix of a graph $G$. Cheeger's inequality for a graph holds:*

$$\frac{\lambda_2}{2} \leq \Phi(G) \leq \sqrt{2\lambda_2}.$$

To improve the conductance lower bound of graph fusion $G_F$, we propose maximizing $f_{CON} = \lambda_2(\mathcal{L})$ by searching appropriate view weights subject to constraints in Eq. (5).

$$\max_w \lambda_2(\sum_{i=1}^{z} w_i L_i). \tag{4}$$

### 3.2.3 OPTIMIZATION SCHEME

To find a multi-view Laplacian $\mathcal{L}$ that maximizes the two objective functions, we need to determine the appropriate weight $w_i$ for each view. As all view weights sum up to 1, the optimization search only needs to consider the first $z - 1$ variables. To ensure meaningful contributions from each input

---

**Algorithm 1:** SMGF

---

**Input:** Graph views $\{G_i\}_{i=1}^a$, attribute views $\{X_i\}_{i=1}^b$, number of classes $k$, algorithm parameters $K, w_{LB}, t, h$.

1   Construct $G_X = \text{KNN}(X, K)$ for attribute views;
2   Compute normalized Laplacian matrices $L_1, \ldots, L_z$;
3   Initialize view weights $w_1, \ldots, w_{z-1} \leftarrow 1/z$ ;
4   Eigengap optimization step: COBYLA $(w_1, \ldots, w_{z-1}, f_{GAP}, \Omega, w_{LB})$ ;
5   Connectivity optimization step: COBYLA $(w_1, \ldots, w_{z-1}, f_{CON}, \Omega, w_{LB}, t)$ ;
6   Multi-view Laplacian $\mathcal{L} \leftarrow w_1 L_1 + \cdots + (1 - w_1 - \cdots - w_{z-1}) L_z$;
7   **if** *embedding* **then**
8       Embedding vectors $u_1, \ldots, u_n \leftarrow \text{NetMF}(\mathcal{L}, h)$;

9   **if** *clustering* **then**
10     Solve the $k$ bottom eigenvectors $y_1, \ldots, y_k \leftarrow \text{eig}(\mathcal{L}, k)$;
11     Clusters $C_1, \ldots, C_k \leftarrow \text{discretize}(y_1, \ldots, y_k)$;

---

view to the graph fusion, we introduce the parameter $w_{LB}$, representing the lower bound of view weights. Consequently, the variables $w_1, \ldots, w_{z-1}$ adhere to the following set of constraints.

$$\Omega(w_1, \ldots, w_{z-1}): \quad w_i \geq w_{LB} \quad \forall \quad 1 \leq i \leq z - 1 \quad \text{and} \quad 1 - \sum_{i=1}^{z-1} w_i \geq w_{LB} \tag{5}$$

Given the difficulty of computing gradients for eigenvalue decomposition, our optimization objectives necessitate using derivative-free constrained optimization techniques. For SMGF, we adopt the COBYLA optimizer, *i.e.*, Constrained Optimization BY Linear Approximation (Powell, 1994). COBYLA iteratively updates a trust region by linear approximations to the objective and constraints.

Both eigengap and connectivity need to be maximized. However, under the assumption $Vol(C) \leq Vol(V)/2$, connectivity is a lower bound of the optimal $k$-way NCut, as opposed to eigengap.

$$\frac{1}{k} \sum_{i=1}^k NCut(C_i) \geq \frac{1}{k} \sum_{i=1}^k \Phi(G) = \Phi(G) \geq \frac{\lambda_2}{2}. \tag{6}$$

A graph composed of $k$ cohesive clusters typically exhibits a low optimal $k$-way normalized cut. Consequently, improving connectivity by maximizing $\lambda_2$ may inadvertently contradict the desired presence of $k$ clusters within $G_F$. Addressing this inherent trade-off between objectives necessitates solving a nontrivial double-objective optimization problem. To address this challenge, we employ a two-step optimization approach (refer to lines 4-5 of Algorithm 1). In the first step, we prioritize the optimization of the eigengap $f_{GAP}$ as our primary objective, which we optimize until convergence. In the second step, we conduct partial optimization of $f_{CON}$ for a specified maximum number of iterations denoted as $t$. This strategy allows us to leverage the connectivity objective as a form of regularization, with the parameter $t$ serving as a means to balance between the two objectives.

### 3.3   Unsupervised learning on $\mathcal{L}$

Utilizing view weights determined through spectrum-guided optimization, we construct the multi-view Laplacian matrix $\mathcal{L}$ as the representation of the graph fusion $G_F$. For deriving node embeddings from graph structure, DeepWalk (Perozzi et al., 2014) is a widely adopted skip-gram model trained on a sampled corpus of random walks. SMGF acquires $h$-dimensional node representations by approximating the training process of DeepWalk via matrix factorization of $\mathcal{L}$, following the NetMF algorithm (Qiu et al., 2018). SMGF acquires $h$-dimensional node representations by factorizing the DeepWalk matrix approximated from $\mathcal{L}$, following NetMF algorithm (Qiu et al., 2018).

Multi-view graph clustering can be achieved through clustering on the underlying graph fusion $G_F$. Consequently, we directly apply spectral clustering to $\mathcal{L}$, which minimizes the normalized cut objective. To extract cluster labels, we employ the Discretize algorithm (Yu & Shi, 2003) on the $k$ eigenvectors of $\mathcal{L}$ associated with $\lambda_1, \ldots, \lambda_k$.

Table 1: Multi-view graph datasets

| Name | $n$ | Graph view (edges) | Attributes $d_X$ | $k$ |
|---|---|---|---|---|
| ACM | 3,025 | Co-author (13,128); Co-label (1,103,868) | 1,870 | 3 |
| DBLP | 4,057 | Co-paper (3,528); Co-conference (2,498,219); Co-term (3,386,139) | 334 | 4 |
| IMDB | 3,550 | Co-director (5,119); Co-actor (31,439) | 2,000 | 3 |
| Amazon Photos | 7,487 | Co-purchase (119,043) | 745; 7,487 | 8 |
| Amazon Computers | 13,381 | Co-purchase (245,778) | 767; 13,381 | 10 |
| MAG | 2,353,996 | Co-author (3,350,128,585); Citation (9,048,278) | 1000 | 22 |

### 3.4 Algorithm analysis

The pseudo-code for SMGF is presented in Algorithm 1. In our time complexity analysis, we treat node degree in graph views, dimension of attribute views, and algorithm hyperparameters as constants. A significant computational bottleneck in SMGF arises from the construction of the KNN graph, which incurs a time complexity of $O(n^2)$. However, the computation and aggregation of single-view Laplacians can be performed in linear time due to their inherent sparsity. The evaluation of eigenvalue-based objectives is accomplished in $O(n)$ time, facilitated by the efficient Arnoldi iterations for sparse matrix eigendecomposition. Similarly, the clustering task in lines 8-9 also exhibits linear time complexity. NetMF carries a time complexity of $O(n^2)$. Therefore, both clustering and embedding on multi-view graphs can be achieved in $O(n^2)$ time using SMGF. For clustering on large-scale data with millions of nodes, we incorporate ScaNN (Guo et al., 2020) for efficient approximate KNN, improving the complexity for clustering to $O(n)$.

## 4 Experiments

In this section, we illustrate the unsupervised learning performance of SMGF through extensive experimentation involving node embedding and clustering tasks on real-world datasets. Additionally, we delve into the impact of alternative objective functions and hyperparameter settings.

### 4.1 Experiment setup

**Datasets.** We perform experimental evaluations using real-world multi-view graph datasets with their respective statistics presented in Table 1. This table provides information on the number of nodes ($n$), specifications of the graph views, dimensions of attribute views, and the count of ground truth node classes ($k$). ACM (Wang et al., 2019), DBLP (Ji et al., 2010), IMDB (Jing et al., 2021), MAG (Sinha et al., 2015) are four datasets that consist of multiple graph views and one attribute view, while Amazon Photos and Amazon Computers (Shchur et al., 2019) are two datasets that each contains one graph view and two sets of node attributes.

**Baselines.** We experimentally compare the performance of our proposed method SMGF against eight baseline algorithms designed for multi-view graphs. For the task of learning node representations, we benchmark our results against four multi-view graph embedding algorithms, namely O2MAC (Fan et al., 2020), DMGI (Park et al., 2020), HDMI (Jing et al., 2021), and URAMN (Zhang et al., 2022). In the context of node clustering, we not only apply K-means to these embedding results but also conduct evaluations of four multi-view graph clustering baselines: MCGC (Pan & Kang, 2021), MvAGC (Lin & Kang, 2021), MVGC (Xia et al., 2022), and MAGC (Lin et al., 2023).

**Evaluation Settings.** To ensure a fair comparison, we test each baseline based on the original implementation and tune hyperparameters accordingly (refer to Appendix A.1 for details). For our approach SMGF, we fix hyperparameters $w_{LB} = 0.05$ and $t = 10$. $K$ in KNN construction is set to 10, except for IMDB dataset where $K = 100$. Embedding dimension $h$ is 64 for all methods.

To report representative performance results, we remove the fixed random seeds in all implementations and repeat 10 runs to get averaged metrics. Experiments are conducted on a Linux computer powered by Intel Xeon 6226R CPU and Nvidia RTX3090 GPU. Note that DMGI, HDMI, and URAMN use GPU, while the other algorithms, including SMGF, are CPU-based. To ensure accurate measurement of running time, we record it in a controlled environment with 16 isolated CPU threads.

Table 2: Node embedding quality for node classification on ACM, DBLP and IMDB (MaF1=Macro-F1, MiF1=Micro-F1). Time in seconds. Best in bold and runner-up underlined.

| | ACM | | | | | DBLP | | | | | IMDB | | | | |
|---|---|---|---|---|---|---|---|---|---|---|---|---|---|---|---|
| Labeled % | 10 | | 50 | | | 10 | | 50 | | | 10 | | 50 | | |
| Metric | MaF1 | MiF1 | MaF1 | MiF1 | Time | MaF1 | MiF1 | MaF1 | MiF1 | Time | MaF1 | MiF1 | MaF1 | MiF1 | Time |
| O2MAC | 0.904 | 0.904 | 0.910 | 0.909 | 112.4 | 0.911 | 0.917 | 0.914 | 0.920 | 679.8 | 0.641 | 0.641 | 0.665 | 0.666 | 672.9 |
| DMGI | 0.780 | 0.795 | 0.908 | 0.909 | 31.25 | 0.920 | 0.926 | 0.924 | 0.929 | 392.1 | 0.658 | 0.658 | 0.676 | 0.676 | 75.94 |
| HDMI | 0.921 | 0.922 | 0.929 | 0.928 | 157.2 | 0.914 | 0.921 | 0.915 | 0.924 | 532.4 | 0.627 | 0.631 | 0.653 | 0.652 | 241.3 |
| URAMN | 0.918 | 0.919 | 0.921 | 0.921 | 59.81 | 0.903 | 0.912 | 0.913 | 0.919 | 153.9 | 0.674 | 0.680 | 0.708 | 0.707 | 124.8 |
| SMGF | 0.927 | 0.927 | 0.933 | 0.932 | 26.40 | 0.926 | 0.932 | 0.931 | 0.936 | 98.04 | 0.690 | 0.690 | 0.724 | 0.724 | 18.84 |

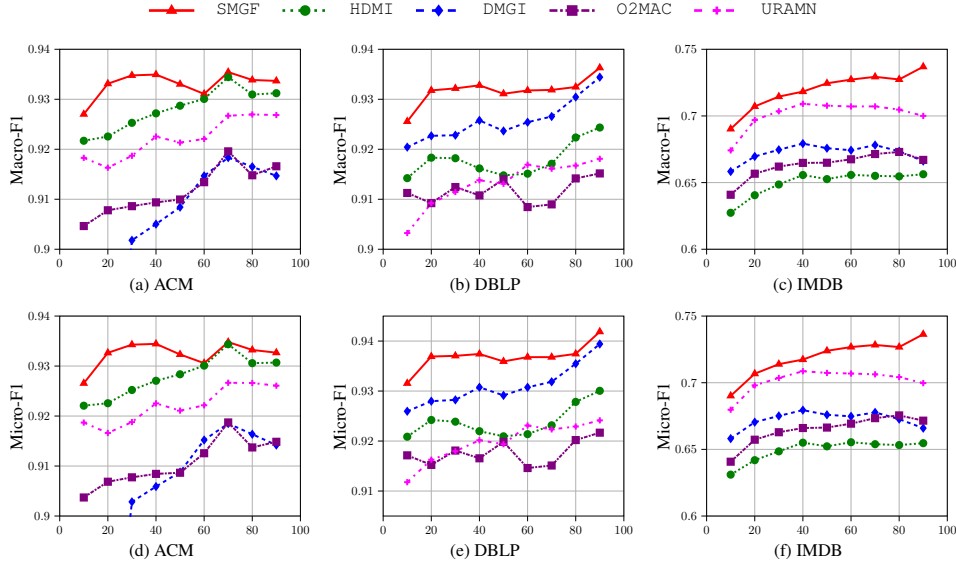

Figure 1: Node classification performance over varying ratio of training data (%).

## 4.2 NODE EMBEDDING EVALUATION

We evaluate the quality of node embeddings through a node classification task. From the acquired node embedding vectors, we train a logistic regression classifier to predict class labels. The results, presented in Table 2, include Macro-F1 (MaF1) and Micro-F1 (MiF1) metrics across varying proportions of training data (Labeled %) in 10% and 50%, as well as embedding time costs in seconds, over the widely used benchmarking classification datasets. Our SMGF consistently exhibits superior embedding quality and efficiency compared to four baseline algorithms designed for multi-view graphs, underscoring its capability for unsupervised representation learning. This is further highlighted in Fig. 1, where SMGF dominates baseline methods over a wide range of labeled ratios.

To visualize the distribution of embedding vectors, we map them to 2-D space using t-SNE. Fig. 2 illustrates the node embeddings of ACM multi-view graph acquired by different methods. Compared with baselines, ACM embeddings acquired by SMGF demonstrate noticeably regular boundaries between the three ground truth classes. Results for other datasets are in Appendix B.4

## 4.3 CLUSTERING PERFORMANCE

**Overall clustering results.** Table 3 reports the clustering performance. Five of the baseline methods are not applicable to datasets with multiple attribute views, *i.e.*, Amazon photos and computers. In both normalized mutual information (NMI) and adjusted Rand index (ARI) metrics, our SMGF achieves the best performance in five out of six datasets. Despite the higher metrics on IMDB, URAMN is much slower than SMGF (Table 5) and requires tuning three hyperparameters for different

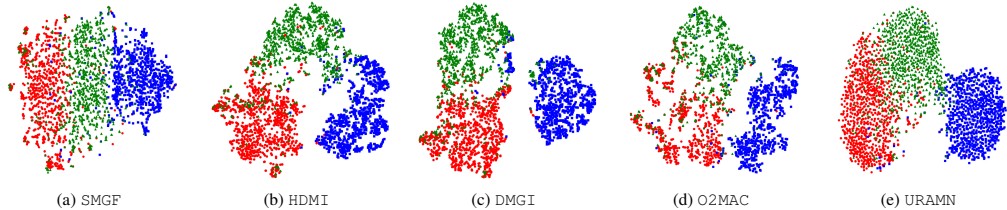

Figure 2: Node embeddings obtained from ACM dataset. Colors represent ground truth classes.

Table 3: Clustering quality with NMI ARI measures. Results marked with $*$ are replicated from the original paper. Best in bold and runner-up underlined.

| Algorithm | ACM NMI | ACM ARI | DBLP NMI | DBLP ARI | IMDB NMI | IMDB ARI | Amazon photos NMI | Amazon photos ARI | Amazon computers NMI | Amazon computers ARI | MAG NMI | MAG ARI |
|---|---|---|---|---|---|---|---|---|---|---|---|---|
| DMGI | 0.703 | 0.747 | 0.732 | 0.790 | 0.197 | 0.200 | - | - | - | - | OOM | |
| HDMI | 0.695 | 0.732 | 0.706 | 0.761 | 0.162 | 0.142 | - | - | - | - | OOM | |
| URAMN | 0.717 | 0.766 | 0.735 | 0.798 | **0.248** | **0.264** | - | - | - | - | OOM | |
| O2MAC | 0.667 | 0.716 | 0.669 | 0.705 | 0.135 | 0.139 | - | - | - | - | OOM | |
| MVGC | 0.645 | 0.641 | $0.742^*$ | $0.804^*$ | 0.118 | 0.126 | - | - | - | - | OOM | |
| MCGC | 0.709 | 0.763 | 0.716 | 0.771 | 0.164 | 0.186 | 0.595 | 0.449 | 0.557 | 0.419 | OOM | |
| MvAGC | 0.603 | 0.636 | 0.650 | 0.708 | 0.191 | 0.201 | 0.558 | 0.384 | 0.512 | 0.365 | 0.049 | 0.004 |
| MAGC | 0.597 | 0.659 | 0.771 | 0.827 | 0.057 | 0.062 | 0.591 | 0.384 | 0.323 | 0.158 | >12h | |
| SMGF | **0.718** | **0.768** | **0.776** | **0.830** | 0.213 | 0.224 | **0.685** | **0.621** | **0.588** | **0.446** | **0.566** | **0.481** |

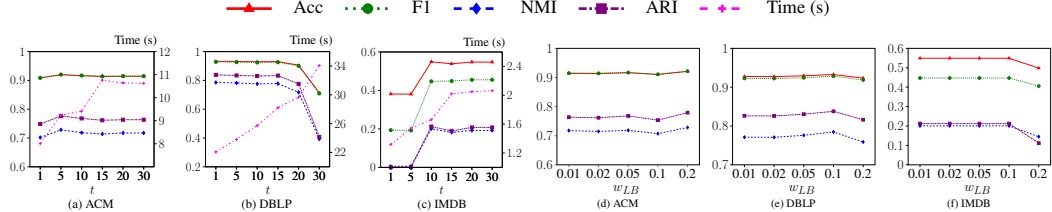

Figure 3: Plots (a)-(c): clustering performance and efficiency with varied parameter $t$. Plots (d)-(f): clustering performance with varied $w_{LB}$.

Table 4: Ablation study on ACM, DBLP, and IMDB dataset

| | ACM NMI | ACM ARI | ACM $w_1$ | ACM $w_2$ | ACM $w_3$ | DBLP NMI | DBLP ARI | DBLP $w_1$ | DBLP $w_2$ | DBLP $w_3$ | DBLP $w_4$ | IMDB NMI | IMDB ARI | IMDB $w_1$ | IMDB $w_2$ | IMDB $w_3$ |
|---|---|---|---|---|---|---|---|---|---|---|---|---|---|---|---|---|
| UNIFORM | 0.611 | 0.577 | 0.33 | 0.33 | 0.33 | 0.756 | 0.815 | 0.25 | 0.25 | 0.25 | 0.25 | 0.007 | 0.002 | 0.33 | 0.33 | 0.33 |
| REG | 0.665 | 0.693 | 0.20 | 0.31 | 0.49 | 0.777 | 0.833 | 0.05 | 0.36 | 0.54 | 0.05 | 0.016 | 0.005 | 0.11 | 0.28 | 0.61 |
| GAP-ONLY | 0.702 | 0.748 | 0.34 | 0.05 | 0.61 | 0.786 | 0.839 | 0.02 | 0.60 | 0.03 | 0.35 | 0.003 | 0.001 | 0.39 | 0.35 | 0.26 |
| CON-ONLY | 0.705 | 0.753 | 0.21 | 0.14 | 0.65 | 0.394 | 0.408 | 0.05 | 0.05 | 0.85 | 0.05 | 0.182 | 0.190 | 0.13 | 0.21 | 0.66 |
| CON-GAP | 0.702 | 0.748 | 0.34 | 0.05 | 0.61 | 0.781 | 0.834 | 0.05 | 0.67 | 0.04 | 0.24 | 0.004 | 0.001 | 0.35 | 0.42 | 0.23 |
| SMGF | 0.718 | 0.768 | 0.22 | 0.16 | 0.62 | 0.776 | 0.830 | 0.11 | 0.50 | 0.34 | 0.05 | 0.213 | 0.224 | 0.19 | 0.26 | 0.54 |

datasets. On the million-scale dataset MAG, most baselines are too slow or run out of memory (OOM), while SMGF discovers high-quality clusters in a reasonable time. Our method also exhibits remarkable efficiency and quality on more metrics (refer to expanded results in Appendix B.1).

**Hyperparameter analysis.** As depicted in Fig. 3, we study the impact of hyperparameters on four clustering metrics. Specifically, we investigate the effects of $w_{LB}$ and $t$ used in our spectral-guided weighting scheme. Fig. 3(b) shows that excessively prioritizing connectivity by setting $t > 20$ has a detrimental impact on DBLP performance. In Fig. 3(c), a deficiency in connectivity also adversely affects clustering quality in the case of IMDB. On the other hand, Fig. 3(d-f) illustrate that SMGF performs consistently well across a range of relatively small values for $w_{LB}$.

**Ablation study.** We perform ablations studies on optimization objectives, to demonstrate the effectiveness of two objective functions and the optimization scheme in Section 3.2. Table 4 compares SMGF against variants with uniform view weighting (UNIFORM), optimizing relative eigengap by Fan et al. (2022) (REG), SMGF without connectivity (GAP-ONLY) or eigen-gap optimization (CON-ONLY), SMGF with reversed optimization steps (GAP-CON). Noticeably, SMGF has the best overall performance. CON-ONLY causes the dense co-term view in DBLP to be predominant in $\mathcal{L}$ and reduces cluster quality. GAP-ONLY underweights the attribute view in IMDB but overly emphasizes the incomplete graph views. These findings underscore the significance of balancing both objectives and provide empirical evidence supporting the efficacy of our optimization scheme.

**Extended experiments.** SMGF also exhibits strong performance when applied to multi-view data comprising solely attribute views, as demonstrated in Appendix B.6. Furthermore, we investigate the choice between Discretize and K-means for clustering in Appendix B.5.

## 5  RELATED WORK

In the domain of multiple graph views, also known as multiplex graphs, previous research on node embedding includes works by Zhang et al. (2018) and Zhang & Kou (2022). For clustering on multiple attribute views, the research dates back to Bickel & Scheffer (2004), and a comprehensive survey is conducted by Fang et al. (2023). A few graph fusion approaches to multi-view clustering have been proposed. Zhou & Burges (2007) aggregate the random walk Laplacians without weighting. Nie et al. (2017) construct a well-clustered graph as the centroid of single-view graphs. Zong et al. (2018) optimize view weights by assuming the consensus result to be close to every single view. Kang et al. (2020) leverages this assumption for graph fusion based on structural graph learning.

For multiple graphs with attributes, a few graph learning methods have been developed. O2MAC, proposed by Fan et al. (2020), learns node embeddings via a graph auto-encoder model with the adoption of reconstruction loss and self-training clustering. Other methods utilize deep graph infomax (Park et al., 2020; Jing et al., 2021) and contrastive learning (Zhang et al., 2022) for multi-view node representation learning. MVGC (Xia et al., 2022) improves O2MAC by introducing attribute augmentation and a new loss function derived from block diagonal constraints. Liu et al. (2022) extend the graph auto-encoder model by incorporating attention mechanism and contrastive fusion.

Other approaches utilize graph filtering techniques to construct a consensus graph. Pan & Kang (2021) leveraged graph filtering and contrastive learning regularization to learn a consensus graph from smoothed node representations. MvAGC (Lin & Kang, 2021) adopts efficient graph filtering and SVD-based spectral clustering leveraging anchor nodes instead of deep learning. MAGC (Lin et al., 2023) exploits higher-order proximity for the optimization of consensus graph.

Results from spectral graph theory (Chung, 1997) have been extensively utilized for graph algorithms, including the spectral clustering algorithm (Shi & Malik, 2000; Ng et al., 2001). A few algorithms have leveraged the spectral graph properties for optimization. Lu et al. (2019) show that minimizing the sum of $k$ smallest eigenvalues of the representation matrix improves the block diagonal property for subspace clustering. Afzalan & Jazizadeh (2019) utilizes the eigengap heuristic to determine the number of clusters automatically. Fan et al. (2022) proposes a relative eigengap objective for automating the choice of hyperparameters in affinity graph construction.

## 6  CONCLUSION

In this paper, we present SMGF, a graph fusion framework that supports unsupervised learning on multi-view graphs with graph views and attribute views. The underlying graph structure among entities is approximated by multi-view Laplacian $\mathcal{L}$, constructed via weighted graph fusion from all views. We formulated two objectives based on eigenvalues of $\mathcal{L}$, motivated by the inherent $k$ classes and graph incompleteness. View weights are determined by a carefully designed two-step optimization scheme. The resulting $\mathcal{L}$ represents a high-quality graph fusion from all views, as evidenced by SMGF's superior embedding and clustering performance.

Looking ahead, we anticipate conducting further investigations into the spectral properties of $\mathcal{L}$ to refine our approach. Additionally, we plan to extend our proposed unsupervised learning framework to accommodate other complex forms of graph data.

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

# A EXPERIMENTAL DETAILS

## A.1 BASELINE EVALUATION

- `DMGI` (Park et al., 2020) We tune the consensus regularization hyperparameter $\alpha$ and l2 regularization coefficient $\beta$ among $\{0.0001, 0.001, 0.01, 0.1\}$ according to their original implementation (`https://github.com/pcy1302/DMGI`). The result with the highest clustering accuracy is reported.

- `URAMN` (Zhang et al., 2022) In accordance with the original hyperparameter specifications at (`https://github.com/RuixZh/URAMN`), we use the authors' settings for the ACM and DBLP datasets. For IMDB, we tune the margin value $\epsilon_1$ and $\epsilon_2$ in $[0.1, 0.9]$, and tune l2 regularization coefficient $\lambda$ among $\{0.0001, 0.001, 0.01, 0.1\}$. We list node classification and clustering results derived from intuitive integrated embedding $\hat{\mathbf{H}}$, which generally outperforms consensus embedding $\mathbf{Z}$ in the original paper and our experiments.

- `HDMI` (Jing et al., 2021): We evaluate this node embedding model based on their original implementation (`https://github.com/baoyujing/HDMI`), which runs `K-means` on embedding vectors for the unsupervised node clustering task. Using identical model configuration for all datasets, `HDMI` saves model parameters with the lowest loss to produce the final embedding vectors.

- `O2MAC` (Fan et al., 2020): Following instructions in the `O2MAC` paper, we train this model for 1000 iterations on DBLP and IMDB datasets, and 250 iterations on ACM dataset. Model implementation (`https://github.com/googlebaba/WWW2020-O2MAC`) and hyperparameter configurations are the same for all datasets. Evaluation is performed on the trained model with the lowest loss value.

- `MVGC` (Xia et al., 2022): Among the attribute augmentations proposed in this work, we evaluate the `MVGC`-Euler variant due to its consistent lead in reported performance. Hyperparameter settings are identical across all datasets, configured as specified in the `MVGC` paper. Their source code is available at `https://github.com/xdweixia/NN-2022-MVGC` but requires substantial modifications to work on datasets other than ACM. The model is trained for 100 epochs and performs clustering every five epochs. Although we observe ACM clustering accuracy over $96\%$ at certain epochs during its training process, the loss function continues to improve afterward while cluster quality declines. Respecting the unsupervised nature of clustering, we record the clustering results at the minimum of the loss function for eventual evaluation.

- `MCGC` (Pan & Kang, 2021): As specified in `MCGC` paper, we fix $m = 2$ and $s = 0.5$ for graph filtering. In the graph learning process, we tune $\alpha$ parameter in the range $\{0.001, 0.1, 1, 10, 100, 1000\}$ on each dataset for best performance and use the $\gamma$ parameter settings in the original implementation (`https://github.com/Panern/MCGC`). Finally, we run the training process to reproduce their reported results.

- `MvAGC` (Lin & Kang, 2021): The balance parameter $\alpha$ and the number of anchors $m$ need to be tuned for each dataset. From the settings of $m$ in the authors' implementation (`https://github.com/sckangz/MvAGC`) and our preliminary experiments, we observe that the number of anchors sufficient for good clustering quality is approximately proportional to the number of nodes in the dataset, so we use $\{n/50-20, n/50, n/50+20\}$ as candidates of $m$. On ACM and DBLP datasets, we use the $\alpha$ candidates in the source code of `MvAGC` and test all combinations of the two hyperparameters. For the other datasets, we first perform a rough search for $\alpha$ over $\{1, 2, 5, 10, 50, 100, 200, 500, 1000\}$ and then narrow it down to a small interval where clustering accuracy is highest. For instance, on Amazon Photos dataset, the highest accuracy found in the first stage occurs at $\alpha = 200$, and we subsequently try $\{120, 150, 180, 200, 220, 250, 300\}$, eventually setting $\alpha = 250$.

- `MAGC` (Lin et al., 2023): The authors of `MAGC` tuned three hyperparameters in their parameter analysis experiments: trade-off parameter $\alpha$, smoothing parameter $\gamma$, and the order of graph filter $k$. In our experiment, $k$ takes value from $\{1, 2, 3, 4, 5\}$, $\alpha$ varies in the range $\{0.1, 1, 2, 5, 10, 100, 1000\}$ while $\gamma$ is fixed to $-1$ as instructed in the `MAGC` paper. On each dataset, we conduct an exhaustive search over all possible hyperparameter combinations by altering configurations in the original implementation (`https://github.com/sckangz/MAGC`) and select the best result based on accuracy value.

Table 5: Clustering performance on ACM, DBLP and IMDB. Mean results on 10 repeats, with standard deviation in brackets. Running time in seconds. Results marked with ∗ are replicated from the original paper. Best in bold and runner-up underlined.

| Algorithm | ACM | | | | | DBLP | | | | | IMDB | | | | |
|---|---|---|---|---|---|---|---|---|---|---|---|---|---|---|---|
| | Acc | F1 | NMI | ARI | Time | Acc | F1 | NMI | ARI | Time | Acc | F1 | NMI | ARI | Time |
| DMGI | 0.907 (0.000) | 0.906 (0.000) | 0.703 (0.000) | 0.747 (0.000) | 34.10 | 0.910 (0.000) | 0.901 (0.000) | 0.732 (0.000) | 0.790 (0.000) | 395.7 | 0.582 (0.000) | 0.576 (0.000) | 0.197 (0.000) | 0.200 (0.000) | 77.35 |
| HDMI | 0.900 (0.004) | 0.899 (0.004) | 0.695 (0.006) | 0.732 (0.008) | 161.2 | 0.895 (0.005) | 0.885 (0.007) | 0.706 (0.009) | 0.761 (0.010) | 537.9 | 0.541 (0.025) | 0.547 (0.026) | 0.162 (0.015) | 0.142 (0.022) | 245.9 |
| URAMN | 0.915 (0.000) | 0.914 (0.000) | 0.717 (0.000) | 0.766 (0.000) | 63.32 | 0.915 (0.000) | 0.910 (0.000) | 0.735 (0.001) | 0.798 (0.001) | 159.5 | 0.670 (0.001) | 0.667 (0.001) | 0.248 (0.001) | 0.264 (0.001) | 129.7 |
| O2MAC | 0.895 (0.006) | 0.897 (0.006) | 0.667 (0.011) | 0.716 (0.014) | 115.0 | 0.873 (0.013) | 0.865 (0.013) | 0.669 (0.024) | 0.705 (0.027) | 684.1 | 0.547 (0.048) | 0.550 (0.047) | 0.135 (0.022) | 0.139 (0.031) | 679.1 |
| MVGC | 0.789 (0.190) | 0.771 (0.217) | 0.645 (0.115) | 0.641 (0.198) | 1631.2 | 0.923∗ (0.000) | 0.923∗ (0.000) | 0.742∗ (0.000) | 0.804∗ (0.000) | 1934.7 | 0.514 (0.009) | 0.499 (0.011) | 0.118 (0.007) | 0.126 (0.009) | 1542.7 |
| MCGC | 0.915 (0.000) | 0.916 (0.000) | 0.709 (0.000) | 0.763 (0.000) | 748.2 | 0.902 (0.000) | 0.895 (0.000) | 0.716 (0.001) | 0.771 (0.001) | 2245.2 | 0.567 (0.001) | 0.545 (0.001) | 0.164 (0.002) | 0.186 (0.003) | 1551.9 |
| MvAGC | 0.861 (0.011) | 0.862 (0.011) | 0.603 (0.024) | 0.636 (0.025) | **4.334** | 0.874 (0.013) | 0.866 (0.013) | 0.650 (0.020) | 0.708 (0.026) | **5.654** | 0.552 (0.007) | 0.469 (0.044) | 0.191 (0.015) | 0.201 (0.015) | 6.399 |
| MAGC | 0.872 (0.000) | 0.872 (0.000) | 0.597 ((0.000) | 0.659 (0.000) | 26.10 | 0.928 (0.000) | 0.923 (0.000) | 0.771 (0.000) | 0.827 (0.000) | 35.98 | 0.484 (0.001) | 0.424 (0.003) | 0.057 (0.001) | 0.062 (0.001) | 33.69 |
| SMGF | **0.916** (0.000) | **0.917** (0.000) | **0.718** ((0.000) | **0.768** (0.000) | 9.080 | **0.929** (0.000) | **0.924** (0.000) | **0.776** (0.000) | **0.830** (0.000) | 26.92 | 0.557 (0.000) | 0.453 (0.000) | 0.213 (0.000) | 0.224 (0.000) | **3.596** |

Table 6: Clustering performance on Amazon photos, Amazon computers and MAG. Mean results on 10 repeats, with standard deviation in brackets. Running time in seconds.

| Algorithm | Amazon photos | | | | | Amazon computers | | | | | MAG | | | | |
|---|---|---|---|---|---|---|---|---|---|---|---|---|---|---|---|
| | Acc | F1 | NMI | ARI | Time | Acc | F1 | NMI | ARI | Time | Acc | F1 | NMI | ARI | Time |
| MCGC | 0.674 (0.001) | 0.582 (0.001) | 0.595 (0.001) | 0.449 (0.001) | 5102.4 | 0.569 (0.001) | **0.531** (0.001) | 0.557 (0.001) | 0.419 (0.001) | 17567.5 | | | OOM | | |
| MvAGC | 0.615 (0.040) | 0.568 (0.032) | 0.558 (0.029) | 0.384 (0.056) | 14.98 | 0.516 (0.015) | 0.426 (0.014) | 0.512 (0.011) | 0.365 (0.014) | 60.20 | 0.261 (0.000) | 0.044 (0.000) | 0.049 (0.000) | 0.004 (0.000) | **4317** |
| MAGC | 0.646 (0.000) | 0.571 (0.000) | 0.591 (0.000) | 0.384 (0.000) | 172.1 | 0.447 (0.000) | 0.348 (0.000) | 0.323 (0.000) | 0.158 (0.001) | 507.5 | | | >12h | | |
| SMGF | **0.787** (0.000) | **0.713** (0.000) | **0.685** (0.000) | **0.621** (0.000) | **7.61** | **0.595** (0.000) | 0.510 (0.000) | **0.588** (0.000) | **0.446** (0.000) | **27.2** | **0.600** (0.000) | **0.335** (0.000) | **0.566** (0.000) | **0.481** (0.000) | 7602 |

# B    EXPANDED EXPERIMENT RESULTS

## B.1    CLUSTERING PERFORMANCE EVALUATION

Tables 5 and 6 present the complete evaluation results of clustering experiments. We include supervised metrics accuracy (Acc) and F1 as well as clustering time costs. There's no randomness in SMGF algorithm, while some baseline algorithms show substantial standard deviations in their results.

## B.2    HYPERPARAMETER ANALYSIS

### B.2.1    VARYING $w_{LB}$, THE LOWER BOUND OF VIEW WEIGHTS

Parameter $w_{LB}$ specifies the lower bound of view weights to ensure that every graph or attribute view of the input multi-view graph contributes to the final clustering results. Our method SMGF sets $w_{LB} = 0.05$ by default. Here we test $w_{LB}$ in $\{0.01, 0.02, 0.05, 0.1, 0.2\}$ and present the results in Fig. 4. Observe that the performance of SMGF remains table from 0.01 to 0.1 on all datasets; the performance increases at 0.2 on ACM and Amazon Photos but decreases on DBLP and IMDB. Therefore, we set $w_{LB} = 0.05$ by default.

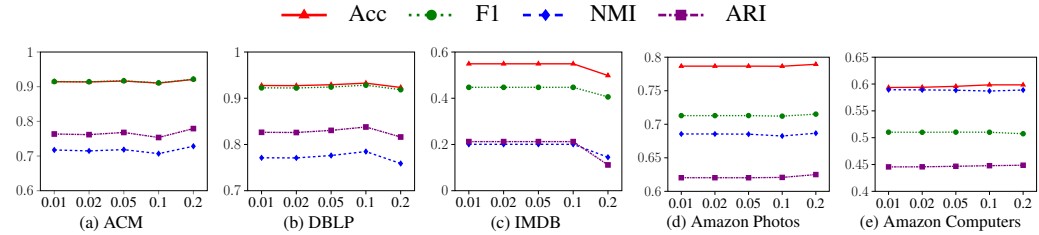

Figure 4: Varying $w_{LB}$.

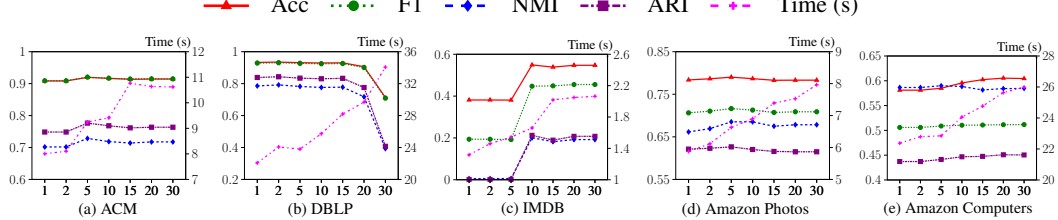

Figure 5: Varying $t$.

Table 7: Ablation study on ACM, DBLP, and IMDB dataset

|  | ACM | | | | | | | DBLP | | | | | | | | IMDB | | | | | | |
|---|---|---|---|---|---|---|---|---|---|---|---|---|---|---|---|---|---|---|---|---|---|---|
|  | Acc | F1 | NMI | ARI | $w_1$ | $w_2$ | $w_3$ | Acc | F1 | NMI | ARI | $w_1$ | $w_2$ | $w_3$ | $w_4$ | Acc | F1 | NMI | ARI | $w_1$ | $w_2$ | $w_3$ |
| UNIFORM | 0.816 | 0.815 | 0.611 | 0.577 | 0.33 | 0.33 | 0.33 | 0.923 | 0.918 | 0.756 | 0.815 | 0.25 | 0.25 | 0.25 | 0.25 | 0.382 | 0.195 | 0.007 | 0.002 | 0.33 | 0.33 | 0.33 |
| REG | 0.884 | 0.886 | 0.665 | 0.693 | 0.20 | 0.31 | 0.49 | 0.930 | 0.926 | 0.777 | 0.833 | 0.05 | 0.36 | 0.54 | 0.05 | 0.389 | 0.215 | 0.016 | 0.005 | 0.11 | 0.28 | 0.61 |
| GAP-ONLY | 0.908 | 0.909 | 0.702 | 0.748 | 0.34 | 0.05 | 0.61 | 0.933 | 0.929 | 0.786 | 0.839 | 0.02 | 0.60 | 0.03 | 0.35 | 0.380 | 0.190 | 0.003 | 0.001 | 0.39 | 0.35 | 0.26 |
| CON-ONLY | 0.910 | 0.910 | 0.705 | 0.753 | 0.21 | 0.14 | 0.65 | 0.714 | 0.709 | 0.394 | 0.408 | 0.05 | 0.05 | 0.85 | 0.05 | 0.538 | 0.448 | 0.182 | 0.190 | 0.13 | 0.21 | 0.66 |
| CON-GAP | 0.908 | 0.908 | 0.702 | 0.748 | 0.34 | 0.05 | 0.61 | 0.931 | 0.926 | 0.781 | 0.834 | 0.05 | 0.67 | 0.04 | 0.24 | 0.381 | 0.192 | 0.004 | 0.001 | 0.35 | 0.42 | 0.23 |
| SMGF | 0.916 | 0.917 | 0.718 | 0.768 | 0.22 | 0.16 | 0.62 | 0.929 | 0.924 | 0.776 | 0.830 | 0.11 | 0.50 | 0.34 | 0.05 | 0.557 | 0.453 | 0.213 | 0.224 | 0.19 | 0.26 | 0.54 |

Table 8: Ablation study on Amazon Photos & Computers dataset

|  | Amazon photos | | | | | | | Amazon Computers | | | | | | |
|---|---|---|---|---|---|---|---|---|---|---|---|---|---|---|
|  | Acc | F1 | NMI | ARI | $w_1$ | $w_2$ | $w_3$ | Acc | F1 | NMI | ARI | $w_1$ | $w_2$ | $w_3$ |
| UNIFORM | 0.789 | 0.717 | 0.686 | 0.623 | 0.33 | 0.33 | 0.33 | 0.605 | 0.515 | 0.578 | 0.427 | 0.33 | 0.33 | 0.33 |
| REG | 0.777 | 0.698 | 0.653 | 0.610 | 0.18 | 0.63 | 0.19 | 0.602 | 0.512 | 0.585 | 0.449 | 0.42 | 0.53 | 0.05 |
| GAP-ONLY | 0.784 | 0.706 | 0.661 | 0.621 | 0.18 | 0.31 | 0.51 | 0.582 | 0.505 | 0.587 | 0.437 | 0.52 | 0.05 | 0.43 |
| CON-ONLY | 0.783 | 0.708 | 0.676 | 0.616 | 0.33 | 0.62 | 0.05 | 0.607 | 0.512 | 0.584 | 0.452 | 0.39 | 0.56 | 0.05 |
| CON-GAP | 0.779 | 0.700 | 0.655 | 0.613 | 0.18 | 0.58 | 0.24 | 0.581 | 0.506 | 0.586 | 0.437 | 0.55 | 0.05 | 0.40 |
| SMGF | 0.787 | 0.713 | 0.685 | 0.621 | 0.36 | 0.56 | 0.08 | 0.595 | 0.510 | 0.588 | 0.446 | 0.47 | 0.43 | 0.10 |

### B.2.2    VARYING CONNECTIVITY REGULARIZATION ITERATIONS $t$

Parameter $t$ is the number of optimization iterations during which we maximize the connectivity objective function $f_{CON}$. SMGF sets $t = 10$ by default. To assess the impact of regularization on performance, we experiment with various settings of $t$ in $\{1, 2, 5, 10, 15, 20, 30\}$. As shown in Fig. 5, the first 10 iterations of connectivity regularization improve cluster quality on all datasets except DBLP. When $t$ reaches 20, we notice a decline in DBLP clustering quality caused by the excessive weight assigned to the highly dense Co-Term graph view. Larger $t$ also leads to a longer execution time. We conclude that setting $t \in [10, 20]$ generally yields favorable performance, and $t = 10$ is a reasonable choice for better efficiency.

### B.3    ABLATION STUDY

We perform ablation studies on SMGF, with expanded results in Tables 7 and 8. UNIFORM is a vanilla version where weights are uniformly assigned to the $z$ views without further optimization. Three alternative methods where COBYLA optimizes a single objective till convergence are also tested, including REG (relative eigengap) (Fan et al., 2022), GAP-ONLY (eigengap objective only,

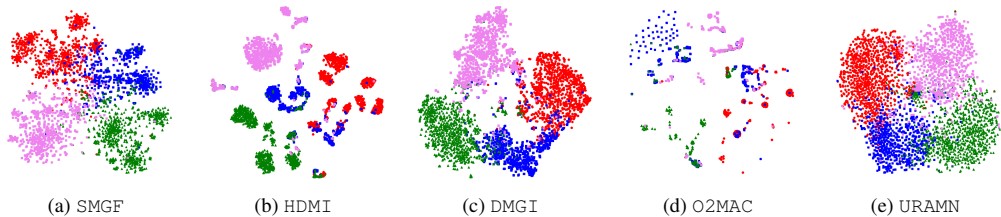

| (a) SMGF | (b) HDMI | (c) DMGI | (d) O2MAC | (e) URAMN |

Figure 6: Visualization of node embeddings obtained from DBLP dataset.

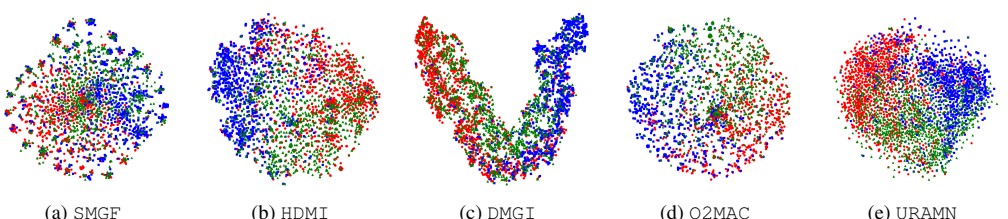

| (a) SMGF | (b) HDMI | (c) DMGI | (d) O2MAC | (e) URAMN |

Figure 7: Visualization of node embeddings obtained from IMDB dataset.

Eq. 3) and CON-ONLY (connectivity objective only, Eq. 4). The maximization objective of relative eigengap is formulated as $f_{reg}(\mathcal{L}) = (\lambda_{k+1}(\mathcal{L}) - \frac{1}{k}\sum_{i=1}^{k}\lambda_i(\mathcal{L}))/(\frac{1}{k}\sum_{i=1}^{k}\lambda_i(\mathcal{L}) + \epsilon)$ , where $\epsilon = 10^{-5}$. Results of the ablation study on five datasets are reported in Tables 7 and 8. In these tables, in addition to Acc, F1, NMI, ARI, and time, the weights of all views in a dataset are also reported. As shown in these tables, on all five datasets, SMGF exhibits superior consistency in performance, which validates the effectiveness of the proposed eigengap and connectivity objectives in Sections 3.2.1 and 3.2.2. Furthermore, as shown in results on DBLP dataset (see Table 7), CON-ONLY without eigengap objective assigns a rather large weight 0.85 to $w_3$; however, the corresponding graph view is quite dense with highly connected nodes. When only the connectivity objective is considered, this view dominates the clustering process with excessive weight. On the other hand, SMGF, with the consideration of both eigengap and connectivity objectives, addresses this issue with a more suitable weight assignment, which demonstrates the effectiveness of the two-stage optimization scheme described in Section 3.2.3.

### B.4 VISUALIZATION OF EMBEDDING VECTORS

For DBLP and IMDB datasets, Fig. 6 and Fig. 7 present the t-SNE visualization of embedding vectors acquired by SMGF and four baseline approaches.

### B.5 DISCRETIZE VS K-MEANS

Spectral clustering is a two-phased algorithm where we first perform eigendecomposition and then calculate discrete labels from eigenvectors. For the second step, SMGF adopts the method proposed by Yu & Shi (2003), referred to as Discretize. K-means is another common alternative in preceding works, and we experimentally evaluate their effectiveness. From the results presented in Table 9, we observe that using Discretize and K-means leads to similarly competitive performance. Nevertheless, K-means occasionally exhibits instability in clustering quality and is consistently less efficient than Discretize.

### B.6 MUTLI-VIEW ATTRIBUTE CLUSTERING

Note that it is not our focus in this paper to handle multi-view data consisting of purely attribute views. Nevertheless, it is indeed possible to adapt SMGF for general multi-view data without any

Table 9: Clustering performance of `K-means` and `SMGF` on five datasets. Mean results on 10 repeats, with standard deviation in brackets. Running time in seconds. Best in bold.

| Algorithm | ACM | | | | | DBLP | | | | | IMDB | | | | |
|---|---|---|---|---|---|---|---|---|---|---|---|---|---|---|---|
| | Acc | F1 | NMI | ARI | Time | Acc | F1 | NMI | ARI | Time | Acc | F1 | NMI | ARI | Time |
| `K-means` | 0.912 (0.000) | 0.912 (0.000) | 0.714 (0.001) | 0.756 (0.001) | 14.189 | 0.928 (0.000) | **0.924** (0.000) | **0.776** (0.000) | 0.827 (0.000) | 35.02 | 0.541 (0.000) | 0.444 (0.000) | 0.194 (0.000) | 0.195 (0.000) | **2.489** |
| `SMGF` | **0.916** (0.000) | **0.917** (0.000) | **0.718** ((0.000) | **0.768** (0.000) | **9.080** | **0.929** (0.000) | 0.924 (0.000) | 0.776 (0.000) | **0.830** (0.000) | **26.92** | **0.557** (0.000) | **0.453** (0.000) | **0.213** (0.000) | **0.224** (0.000) | 3.596 |

| Algorithm | Amazon photos | | | | | Amazon computers | | | | |
|---|---|---|---|---|---|---|---|---|---|---|
| | Acc | F1 | NMI | ARI | Time | Acc | F1 | NMI | ARI | Time |
| `K-means` | 0.751 (0.003) | 0.664 (0.002) | 0.633 (0.003) | 0.531 (0.009) | 9.683 | **0.597** (0.047) | **0.530** (0.005) | 0.551 (0.002) | 0.359 (0.036) | 36.82 |
| `SMGF` | **0.787** (0.000) | **0.713** (0.000) | **0.685** (0.000) | **0.621** (0.000) | **7.61** | 0.595 (0.000) | 0.510 (0.000) | **0.588** (0.000) | **0.446** (0.000) | **27.2** |

Table 10: Multi-view attribute datasets

| Name | $n$ | Attributes ($d_X$) | $k$ |
|---|---|---|---|
| Yale | 165 | Intensity (4,096), LBP (3,304), Gabor (6,750) | 15 |
| ALOR | 10,800 | Similarity (77), Haralick (13), HSV (64), RGB (125) | 100 |
| NUS-WIDE | 30,000 | CH (64), CM (225), CORR (144), EDH (73), WT(128) | 31 |

Table 11: Clustering performance for multi-view attribute data.

| Algorithm | Yale | | | ALOI | | | NUS-WIDE | | |
|---|---|---|---|---|---|---|---|---|---|
| | ACC | F1 | NMI | ACC | F1 | NMI | ACC | F1 | NMI |
| `SwMC`(Nie et al., 2017) | 0.612 | 0.663 | 0.448 | OOM | | | OOM | | |
| $\text{M}^2\text{VEC}_{km}$(Tao et al., 2020) | 0.452 | 0.485 | 0.204 | OOM | | | OOM | | |
| $\text{M}^2\text{VEC}_{spec}$(Tao et al., 2020) | 0.440 | 0.501 | 0.236 | OOM | | | OOM | | |
| `LMVSC`(Kang et al., 2019) | 0.552 | 0.616 | 0.373 | 0.566 | 0.750 | 0.450 | 0.121 | 0.086 | 0.024 |
| `SMVSC`(Sun et al., 2021) | 0.570 | 0.617 | 0.376 | 0.343 | 0.574 | 0.170 | 0.178 | 0.114 | 0.053 |
| `FPMVS-CAG`(Wang et al., 2021) | 0.442 | 0.498 | 0.253 7 | 0.315 | 0.555 | 0.161 | **0.194** | 0.123 | **0.064** |
| `FastMICE`Huang et al. (2023) | **0.642** | 0.679 | 0.467 | 0.756 | 0.834 | 0.664 | 0.144 | 0.134 | 0.051 |
| `SMGF` | 0.636 | **0.683** | **0.492** | **0.808** | **0.855** | **0.689** | 0.151 | **0.152** | 0.059 |

predefined graph structure. Specifically, `SMGF` handles such data by transforming each attribute view to the normalized Laplacian of its KNN graph.

We perform additional experiments on 3 datasets: Yale, ALOI, and NUS-WIDE, with statistics and experimental results provided in Table 10 and 11. It is remarkable to observe that our `SMGF` is powerful enough to achieve comparable or outstanding performance on these multi-view datasets, compared with existing methods. The evaluation results of baseline algorithms are transcribed from Huang et al. (2023).

## B.7 VISUALIZATION OF OPTIMIZATION PROCESS

In this section, we closely examine the optimization process of eigen-gap and connectivity objectives in `SMGF`. Specifically, the goal is to find out whether the first optimization step maximizes the eigen-gap and how much it deteriorates when connectivity is maximized in the next step.

To demonstrate the distribution of two objectives over the constrained space of weight parameters, we sample view weight combinations via an exhaustive grid search at an interval of 0.05. The corresponding objective values are plotted as the $x/y$ coordinates of dots in Fig. 8. As the eigen-gap is optimized (Stage 1), the red line is the locus of objectives corresponding to the weight parameters updated in each iteration. On every dataset, we observe that view weights converge towards a local

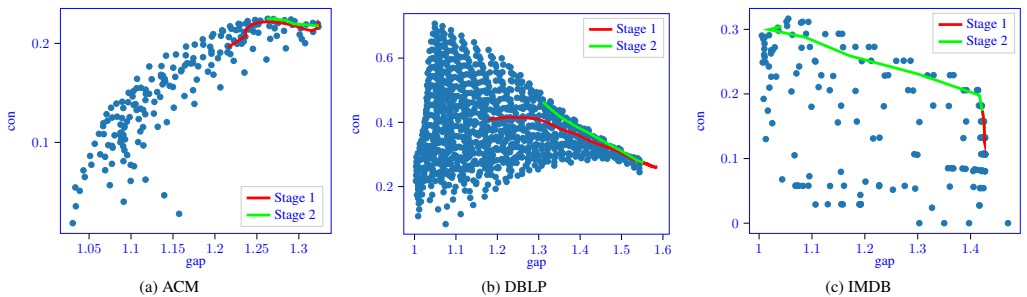

Figure 8: Visualization of the optimization process of eigen-gap and connectivity objectives.

optimum of eigengap (on IMDB, the trajectory is directed downwards). This implies a positive answer to our first question and suggests a continued manifold exists for the eigengap.

The green line that starts from the endpoint of Stage 1 illustrates the trajectory of connectivity optimization. On every dataset, this locus moves towards the top-left direction, which indicates a trade-off between two objectives. We also notice that the green line extends along the boundary of dots, which suggests that the update of view weights approximately traverses the Pareto front of the eigengap and connectivity objectives. In other words, the eigengap is maintained as large as possible while improving connectivity. These results validate that SMGF optimizes both objectives and balances them.

