# OpenReview forum: "Spectrum-guided Multi-view Graph Fusion"
_ICLR.cc/2024/Conference — Submitted to ICLR 2024_

### Official Review · Reviewer_Yebg · 2023-10-24

**Soundness:** 2 fair
**Presentation:** 2 fair
**Contribution:** 2 fair
**Rating:** 5
**Confidence:** 3

**Summary:**

This paper presents SMGF, a novel graph fusion framework that approximates underlying entity connections by aggregating view-specific graph structures. The authors construct a multi-view Laplacian L from normalized Laplacian matrices representing all views. View weights are determined through the optimization of two objectives derived from L’s spectral properties, which exploit the eigenvalue gap and enhance connectivity. Comprehensive experiments on six real-world datasets showcase the superior performance of SMGF in node embedding and clustering results, along with its efficiency and scalability

**Strengths:**

1. The originality, quality, and significance is supported by the proposed SMGF, which is addresses performance and interpretability with a graph fusion framework.

2. The clarity of this paper is satisfied based on the clearly presented motivations and contributions in Introduction part and good organizations in Methodology part.

**Weaknesses:**

1. The biggest problem of this paper is the limited novelty of the formulation in introducing a weighted graph fusion mechanism that directly aggregates the single-view Laplacians. The rationality of the weighted graph fusion is not clear based on the related parts in this paper.

2. The Theorem 1 and Theorem 2 are built on the existing works as the cited works. Then directly placing them on the methodology part will more or less limit the novelty of this paper. I think placing them on the Section 2 will be better and the authors can add more explanation of rationality of the proposed method.

3. What are the major differences between the proposed SMGF and the most related works? I think the authors can add more analysis and comparison in Introduction part. Then the novelty of this paper is more clear.

4. In the experimental part, the authors can add more descriptions of experimental settings in validating the proposed SMGF. Then the authors are easily follow up this work.

5. The authors can add more datasets with large scales and compared methods in the experiment to better demonstrate the effectiveness of the proposed method.

6. A '.' should be added in the end of an equation, i.e., Eq. (4) and Eq. (6).

**Questions:**

I wonder whether there exist relations between eigengap and connectivity as shown in 3.2.1 and 3.2.2.

---

> ### Author Response · Authors · 2023-11-16
>
> Thanks for your review.
>
> **W1:** The biggest problem of this paper is the limited novelty of the formulation in introducing a weighted graph fusion mechanism that directly aggregates the single-view Laplacians. The rationality of the weighted graph fusion is not clear based on the related parts in this paper.
>
> **Response:** Although the weighted graph fusion is a classic methodology, the key point is how to derive the weights effectively and efficiently. Our novel contributions include the nontrivial design of the two objective functions and the efficient optimization algorithm to get the weights, as well as the handling of graph views and attribute KNN graphs in one graph fusion problem in our work.
>
> In Section 3.1, we hypothesize that the true underlying relationships
> among entities can be considered as a certain combination of these individual views. Some of the views may be of a higher significance, and should thus be weighted appropriately. We will also provide an analysis of graph fusion from the perspective of minimizing multi-view normalized cut.
>
> **W2:** The Theorem 1 and Theorem 2 are built on the existing works as the cited works. Then directly placing them on the methodology part will more or less limit the novelty of this paper. I think placing them on the Section 2 will be better and the authors can add more explanation of rationality of the proposed method.
>
> **Response:** Thanks for the suggestion. We consider moving these two theorems to Preliminaries section and reorganizing the methodology section.
>
> **W3:** What are the major differences between the proposed SMGF and the most related works? I think the authors can add more analysis and comparison in Introduction part. Then the novelty of this paper is more clear.
>
> **Response:** Compared to previous graph fusion methods, our design of objective functions for determining view weights is a novel contribution. We list a few notable works on multi-view weighted graph fusion and their objectives.
>
> | Algorithm | Reference | Objective function |
> | -------- | -------- | -------- |
> | PwMC (2017)   | [1] | $min_{w,S}\sum_i^z w_i\|\|S-A_i\|\|_F +\|\|w\|\|_2^2$    |
> | SwMC (2017)   | [1] | $min_S\sum_i^z w_i\|\|S-A_i\|\|_F$ ; $w_i=\frac{1}{2\|\|S-A_i\|\|_F}$  |
> | WMSC (2018)   | [2] | $min_w\sum_i^z \|\|L^*V_i-V_i\Lambda_i\|\|_F+\alpha w^T Qw+\beta w^T w$    |
> | GFSC (2020)   | [3] | $min_{Z,S}\sum_i^z (\|\|X_i-X_iZ_i\|\|_F+\alpha \|\|Z_i\|\|_F+\beta w_i\|\|Z_i-S\|\|_F)$  ; $w_i=\frac{1}{2\|\|Z_i-S\|\|_F}$    |
> | CoALa (2021)   | [4] | $w_i=\frac{1}{4}\beta(2-\lambda_2(L_i))(Silhouette(eigv_2(L_i))+1)$   |
> | MvAGC (2021)   | [5] | $min_{w,S}\sum_i^z w_i(\|\|X_i^T-X_i^TMM^TS\|\|_F+\alpha \|\|S-f(A_i)M\|\|_F+w_i^\gamma)$  |
> | MAGC (2023)   | [6] | $min_{w,S}\sum_i^z w_i(\|\|X_i^T-X_i^TS\|\|_F+\alpha \|\|S-f(A_i)\|\|_F+w_i^\gamma)$  |
> | SMGF (Ours)   |  | $max_w \frac{\lambda_{k+1}(L^*)}{\lambda_{k}(L^*)}$ and $max_w \lambda_2(L^*)$   |
>
> Among the previous view weighting methods, most of them explicitly constrains the fused graph (adjacency $S$, Laplacian $L^*$) to approximate each view (adjacency $A_i$ or $Z_i$, attributes $X_i$ or eigenspace $V_i$). CoALa adopts simple heuristics based on the second eigenvalue and Silhouette index derived from the second eigenvector. In contrast, our objective function for weighting exclusively promotes two spectral properties of the graph fusion represented by $L^*$. This approach is more robust against irregular or incomplete views (as in ACM/IMDB datasets).
>
> Our objective proves highly effective in experiments. PwMC and SwMC have been compared in MAGC and MvAGC papers, all of which are outperformed by SMGF in clustering performance. The other methods (WMSC, GFSC, CoALa) handle the similarity graphs or purely numerical attribute data, and are thus not readily comparable to our task.
>
> [1] Self-weighted Multiview Clustering with Multiple Graphs. IJCAI 2017.
>
> [2] Weighted Multi-View Spectral Clustering Based on Spectral Perturbation. AAAI 2018.
>
> [3] Multi-graph fusion for multi-view spectral clustering. Knowledge-Based Systems 2020.
>
> [4] Approximate Graph Laplacians for Multimodal Data Clustering. TPAMI 2021.
>
> [5] Graph Filter-based Multi-view Attributed Graph Clustering. IJCAI 2021.
>
> [6] Multi-View Attributed Graph Clustering. TKDE 2023.

---

> ### Author Response · Authors · 2023-11-16
>
> (continued)
>
> **W4:** In the experimental part, the authors can add more descriptions of experimental settings in validating the proposed SMGF. Then the authors are easily follow up this work.
>
> **Response:** Parameter settings and system specifications are given in Section 4.1 Evaluation Settings. The source code of our implementation is also provided in the repository (link in abstract). We will further refine the usage instructions in the code repository upon public release.
>
> **W5:** The authors can add more datasets with large scales and compared methods in the experiment to better demonstrate the effectiveness of the proposed method.
>
> **Response:** In addition to the previously tested MAG dataset consisting of publications in maths and physics, we've extracted two new multi-view graphs from other research domains in the Microsoft Academic Graph, named MAG-Eng and MAG-Soc. Both datasets consist of around 1-2 million nodes ($n$) and over 50 ground truth clusters ($k$).
>
> | Dataset | $n$       | $k$ | Algorithm | NMI   | ARI   | Time (s) | RAM (GB) |
> | ------- | --------- | --- | --------- | ----- | ----- | -------- | -------- |
> | MAG-Eng | 1,798,717 | 55  | MvAGC     | 0.230 | 0.017 | 3181.4   | 85.9     |
> |         |           |     | SMGF      | 0.474 | 0.268 | 7541.3   | 44.4     |
> | MAG-Soc | 849,087   | 51  | MvAGC     | 0.023 | 0.015 | 427.7    | 32.7     |
> |         |           |     | SMGF      | 0.482 | 0.195 | 2001.7   | 21.2     |
>
> The remaining baselines continue to fail due to running over 12 hours (MAGC) or out of memory (others). Our SMGF algorithm consistently demonstrates unmatched clustering performance on these large-scale datasets. Despite being slower than MvAGC, SMGF achieves much higher NMI and ARI metrics and is more efficient in RAM space.
>
> **W6:** A '.' should be added in the end of an equation, i.e., Eq. (4) and Eq. (6).
>
> **Response:** Thanks for pointing this out. We will fix it in the updated version.
>
> **Q1:** I wonder whether there exist relations between eigengap and connectivity as shown in 3.2.1 and 3.2.2.
>
> **Response:** The numerical values of eigengap and connectivity are not directly associated. In Section 3.2.3, we show that the connectivity objective value gives a lower bound of the optimal $k$-way NCut. As eigengap optimization is aimed to suppress the upper bound of cluster's NCut, an excessively large connectivity contradicts the motivation of promoting eigengap. Therefore, the optimization scheme in SMGF is designed to reach a balance between both objectives, which could be adjusted by the $t$ parameter controlling the extent of connectivity optimization.

---

### Official Review · Reviewer_zn5a · 2023-10-27

**Soundness:** 2 fair
**Presentation:** 3 good
**Contribution:** 2 fair
**Rating:** 3
**Confidence:** 5

**Summary:**

This paper proposes  a multi-view graph fusion method which can be used in representation learning and clustering. It designs an eigengap objective and a connectivity objective for Laplacian learning. The exeperimental results demonstrate the effectiveness.

**Strengths:**

1. The multi-view graph learning setting is interesting.
2. The experimental results are good.

**Weaknesses:**

1. Although the multi-view graph fusion problem is interesting, the method to tackle this problem is somewhat straightforward. It directly contruct the graph for each attribute view and transfer the problem to the conventional multiple graph fusion setting. Therefore, the proposed method is still a multiple graph fusion method essentially and do not provide any deeper insight to the multi-view graph fusion problem. After constructing the Laplacian matrix for each attribute view, the method only uses the Laplacian without the attribute view, and thus the rich information behind the attribute view is abandoned. However, I think it is the information in the multiple attribute view that is the key difference between the multi-view graph learning and the conventional multiple graph learning.

2. Since the method uses multiple Laplacian, whose size is n-by-n, for fusion. The space complexity should also be provided. In the experiments, the authors use the MAG data set, which is a large scale data set. Intuitively, the proposed method may suffer from the out-of-memoty problem on this data set, but the experimental results show that the proposed one can run a result. It would be better to explain why and how the proposed method can handle this large scale data set. Could you report the memory consumed by the proposed method?

3. The optimization is a two-step method, i.e., it first optimize the eigengap objective and then it optimizes the connectivity objective. I think it is a suboptimal and need to be justified. For example, in the second step, when optimizing the eigengap objective, it is very probably that the eigengap objective will increase a lot. How to balance these two objectives?

4. It would be better to provide the time complexity of the COBYLA algorithm in Lines 4 and 5 in Algorithm 1.

**Questions:**

See above.

---

> ### Author Response · Authors · 2023-11-16
>
> Thanks for your review.
>
> **W1:** Although the multi-view graph fusion problem is interesting, the method to tackle this problem is somewhat straightforward. It directly contruct the graph for each attribute view and transfer the problem to the conventional multiple graph fusion setting. Therefore, the proposed method is still a multiple graph fusion method essentially and do not provide any deeper insight to the multi-view graph fusion problem. After constructing the Laplacian matrix for each attribute view, the method only uses the Laplacian without the attribute view, and thus the rich information behind the attribute view is abandoned. However, I think it is the information in the multiple attribute view that is the key difference between the multi-view graph learning and the conventional multiple graph learning.
>
> **Response:** We consider the simplicity and effectiveness of our approach as a strength that should be appreciated. The aggregation of single-view graphs and the objective functions are simple but effective. Such simplicity allows us to see this graph fusion problem from a much clearer perspective: regardless of the details in each view, what is desired for a good graph fusion itself? This guides the design of two objectives based on the spectrum of multi-view Laplacian. It turns out that our simplistic approach outperforms deep learning based methods as validated by experiments. For future work, our approach also provides a framework to which alternative objectives can be introduced.
>
> We acknowledge that deep learning models have the potential to capture latent information from attribute data. In the datasets used in this work, node attributes are preprocessed binary word vectors or TF-IDF vectors. Under such typical scenario, our results show that the linear proximity relations in KNN suffice for clustering and embedding tasks. Our method is also much more efficient and scalable than deep learning baselines.
>
> **W2:** Since the method uses multiple Laplacian, whose size is n-by-n, for fusion. The space complexity should also be provided. In the experiments, the authors use the MAG data set, which is a large scale data set. Intuitively, the proposed method may suffer from the out-of-memory problem on this data set, but the experimental results show that the proposed one can run a result. It would be better to explain why and how the proposed method can handle this large scale data set. Could you report the memory consumed by the proposed method?
>
> **Response:** SMGF has a peak RAM usage of 57.3GB on MAG dataset, compared to 201.6 GB of MvAGC. We adopt the approximate KNN algorithm ScaNN, which is efficient but is also the memory bottleneck. As the co-author relations are inferred from the paper-author bipartite-graph with adjacency matrix $H$ (with 17M nonzero entries), the dense co-author graph view can be represented by $HH^T$. Therefore, we construct the multi-view Laplacian as a scipy.sparse linear operator, instead of instantiating the full $n\times n$ matrix. Nevertheless, even if SMGF directly takes the graph $HH^T$ as input, the algorithm can still finish in 5.3 hours with 63GB RAM consumption.
>
> **W3:** The optimization is a two-step method, i.e., it first optimize the eigengap objective and then it optimizes the connectivity objective. I think it is a suboptimal and need to be justified. For example, in the second step, when optimizing the eigengap objective, it is very probably that the eigengap objective will increase a lot. How to balance these two objectives?
>
> **Response:** As we first optimize the eigengap objective till convergence, the optimization process of the second objective typically follows the Pareto front of the two objectives. This is validated in additional experiment results in Appendix B.7, where dots represent the grid-search result of objective values under all view weight combinations. The optimization of connectivity does sacrifices the eigengap, which is an inevitable tradeoff.
>
> It's true that a balance between these two objectives is necessary for the optimal performance. With performance and efficiency both considered, the proposed two-step optimization scheme can find a middle point between the optimum of two objectives. The balance between them could be adjust explicitly via the number of optimization iterations $t$. As shown in Figure 5, the performance of SMGF is consistent for $t$ within [10, 20].
>
> **W4:** It would be better to provide the time complexity of the COBYLA algorithm in Lines 4 and 5 in Algorithm 1.
>
> **Response:** COBYLA optimizer returns the next set of weight parameters to be tested based on the past trials of weights. The complexity of COBYLA itself is constant with regard to the size of dataset, while evaluation of eigenvalue objectives requires O(n) time.

---

### Official Review · Reviewer_DRP3 · 2023-11-01

**Soundness:** 2 fair
**Presentation:** 3 good
**Contribution:** 2 fair
**Rating:** 3
**Confidence:** 4

**Summary:**

The paper introduces SMGF, a novel framework for graph fusion in the context of multi-view graphs, which aim to capture diverse relations among entities and individual characteristics through both graph views and attribute views. The primary motivation for this work is the limitations observed in existing approaches, including issues related to efficacy, efficiency, and explicit control over view contributions. SMGF approximates the connections between entities by aggregating view-specific graph structures and constructs a multi-view Laplacian matrix from normalized Laplacian matrices representing all views. View weights are determined through the optimization of two objectives based on the spectral properties of the Laplacian matrix, leveraging eigenvalue gap and connectivity enhancement. The paper presents comprehensive experiments conducted on six real-world datasets, demonstrating that SMGF outperforms existing methods in terms of node embedding, clustering results, efficiency, and scalability.

**Strengths:**

1.The motivation is clear
2. The paper is well written and well structured

**Weaknesses:**

1. The first issue is that the topic of graph fusion in multi-view settings is relatively outdated. Currently, the explanation and performance of existing approaches are quite satisfactory. The motivation stated in this paper, "Existing approaches often lack efficacy, efficiency, and the ability to explicitly control view contributions," is not valid. There are various methods for weighting different views and plenty of adaptive weighting techniques. The algorithms also exhibit linear time complexity. Therefore, I find this motivation less convincing.

2. The second concern is the use of spectral properties based on spectral clustering, which doesn't appear to be a novel contribution. The method section mainly revolves around basic concepts, and I believe the essence of the graph fusion process is not adequately explained. It remains unclear why a particular view dominates, and the approach based on "eigengap and connectivity objectives" appears rather ordinary.

3. The third issue is the mediocre performance of the algorithm. It seems that the improvement in algorithm performance is minimal, and the paper lacks statistical analysis to support the claimed enhancements. Additionally, there is no significant analysis demonstrating the effectiveness. Moreover, the algorithm's performance on the IMDB dataset is considerably worse compared to URAMN.

**Questions:**

N/A

---

> ### Author Response · Authors · 2023-11-16
>
> Thanks for your review.
>
> **W1:** The first issue is that the topic of graph fusion in multi-view settings is relatively outdated. Currently, the explanation and performance of existing approaches are quite satisfactory. The motivation stated in this paper, "Existing approaches often lack efficacy, efficiency, and the ability to explicitly control view contributions," is not valid. There are various methods for weighting different views and plenty of adaptive weighting techniques. The algorithms also exhibit linear time complexity. Therefore, I find this motivation less convincing.
>
> **Response:** By the quoted sentence, we mean that existing approaches often underperform in at least one of the three aspects, namely effectiveness, efficiency and interpretability. The following comparisons highlight that no previous approach combines these strengths like our method SMGF does.
>
> * On all 3 datasets tested for node embedding task, SMGF claims the highest quality and the highest efficiency, as shown in Table 4.
> * In the following Table A, baseline methods (URAMN, MAGC) with comparable clustering quality are always slower than SMGF. MAGC has close efficiency on DBLP, but it is on average 0.179 behind SMGF in the other NMI/ARI metrics (and does not scale for MAG). No baseline method achieves SMGF's level of effectiveness and efficiency simultaneously.
>
> Table A
>
> | Dataset | Ours | Time (s)  | Baseline with best NMI/ARI | Time (s)  | Speedup |
> | ------- | ---- | ----- | ------------- | ----- | ------------ |
> | ACM     | SMGF | 9.080 | URAMN         | 63.32 | 7.0x faster  |
> | DBLP    | SMGF | 26.92 | MAGC          | 35.98 | 1.3x faster  |
> | IMDB    | SMGF | 3.596 | URAMN         | 129.7 | 36.1x faster |
>
> * MvAGC is a faster clustering algorithm than SMGF, but its clustering quality is not on par with our approach, as shown in Table 3.
> * Among the tested baselines, MCGC, MAGC and MvAGC explicitly determines a weight parameter for each view. The remaining methods do not give explicit indicator of each view's contribution to the outcome.
>
> Other previous works on multi-view graph fusion mostly tackle the multi-view clustering problem, where graphs are constructed from attributes via KNN or kernel functions. Our method deals with incomplete and skewed real-world networks by our design of weighting objectives, which focus on the fused multi-view Laplacian and explicitly promotes connectivity. (Please see also our response to Reviewer Yebg W3, where we compare a few objectives from previous graph fusion methods.) We would appreciate if you provide additional references to algorithms that should be analyzed and compared with our approach.
>
>
> **W2:** The second concern is the use of spectral properties based on spectral clustering, which doesn't appear to be a novel contribution. The method section mainly revolves around basic concepts, and I believe the essence of the graph fusion process is not adequately explained. It remains unclear why a particular view dominates, and the approach based on "eigengap and connectivity objectives" appears rather ordinary.
>
> **Response:** Exploiting these spectral properties for multi-view graph learning is a novelty of our work. Previous research on multi-view graph fusion has leveraged the closeness of subspace between each view and the graph fusion [1, 2], while we find the eigenvalues of the multi-view graph Laplacian itself could guide view weighting effectively.
>
> In SMGF algorithm, view weights are updated so that the two objectives are optimized. From the results in Tables 7-8, promoting connectivity objective can make a highly connected view become overly prevalent, as in the case of DBLP's co-term view $G_3$ (this densest view is weighted 0.85 by SMGF's connectivity-only variant). On the other hand, improving eigengap usually leads to a combination of 2-3 views, which implies that cluster structure is related to multiple views.
>
> These two eigenvalue-based objectives cannot be optimized via derivative-based methods such as gradient descent, and we find COBYLA to be a suitable tool for derivative-free optimization. Previous work [3] on automated spectral clustering maximizes eigengap by grid search or Bayesian optimization (which is slower than grid search). COBYLA only requires dozens of objective evaluations in our method, which is much more efficient than grid search.
>
> [1] Weighted Multi-View Spectral Clustering Based on Spectral Perturbation. AAAI 2018.
>
> [2] Approximate Graph Laplacians for Multimodal Data Clustering. TPAMI 2021.
>
> [3] A Simple Approach to Automated Spectral Clustering. NIPS 2022.

---

> ### Author Response · Authors · 2023-11-16
>
> (continued)
>
> **W3:** The third issue is the mediocre performance of the algorithm. It seems that the improvement in algorithm performance is minimal, and the paper lacks statistical analysis to support the claimed enhancements. Additionally, there is no significant analysis demonstrating the effectiveness. Moreover, the algorithm's performance on the IMDB dataset is considerably worse compared to URAMN.
>
> **Response:** On ACM, DBLP and IMDB datasets where SMGF and all 8 baselines can be tested, we calculate the average ranking of all metrics in Tables 2-3. SMGF has the best Avg Rank among all methods, which demonstrates the excellent effectiveness and consistency of our approach.
>
> * Embedding rankings
>
> | Algorithm | Avg Rank (12 F1 metrics) |
> |-----------|-------------|
> | O2MAC     | 4           |
> | DMGI      | 3.3         |
> | HDMI      | 3.3         |
> | URAMN     | 3.3         |
> | SMGF      | 1           |
>
> SMGF ranks 1st in all embedding quality metrics.
> * Clustering rankings
>
> | Algorithm | Avg Rank (6 NMI/ARI metrics) |
> |-----------|----------|
> | DMGI      | 4.2      |
> | HDMI      | 6        |
> | URAMN     | 2.3      |
> | O2MAC     | 7.2      |
> | MVGC      | 6.2      |
> | MCGC      | 4.7      |
> | MvAGC     | 6.8      |
> | MAGC      | 6.3      |
> | SMGF      | 1.3      |
>
> SMGF ranks 1st in 10 clustering NMI/ARI metrics out of 12.
>
> Moreover, on Amazon computers/photos datasets, the NMI and ARI metrics of SMGF surpass the best competitor by 0.09/0.03 and 0.17/0.03. The only baseline method (MvAGC) that can finish on MAG dataset has NMI/ARI 0.049/0.004, while ours achieves 0.566/0.481. Results on all these diverse datasets validates the outstanding effectiveness of SMGF.
>
> On the specific case of IMDB dataset, despite its stronger clustering performance, URAMN's node embeddings consistently fall behind SMGF in classification performance, as shown in Table 2 and Figure 1. Thus, we find the claim that "SMGF's performance on the IMDB dataset is considerably worse compared to URAMN" to be ungrounded.

---

### Official Review · Reviewer_nNdD · 2023-11-01

**Soundness:** 2 fair
**Presentation:** 2 fair
**Contribution:** 2 fair
**Rating:** 5
**Confidence:** 3

**Summary:**

This paper proposed a novel graph fusion framework that approximates underlying entity connections by aggregating view-specific graph
structures, which constructs a multi-view Laplacian L from normalized Laplacian matrices representing all views. Comprehensive experiments on six real-world datasets showcase the superior performance of the proposed method in node embedding and clustering results, along with its efficiency and scalability.

**Strengths:**

1. The proposed method has gained performance improvement when compared to previous multi-iew clustering methods;
2. The time complexity of the proposed algorithm is O(n^2), which is efficient.

**Weaknesses:**

1. For attribute views, how to construct G_X? Since X has multiple views, how many G_x should be constructed?
2. The presentation of the paper is confusion, I cannot see what is the final objective function.

**Questions:**

See weaknesses.

---

> ### Author Response · Authors · 2023-11-16
>
> Thanks for your review.
>
>
> **W1:** For attribute views, how to construct G_X? Since X has multiple views, how many G_x should be constructed?
>
> **Response:** An attribute view $X$ consists of $n$ attribute vectors $x_1, ..., x_n$. The corresponding $G_X$ is a K-nearest neighbor graph where each node $v_i$ represents the attribute vector $x_i$. A multi-view graph may contain multiple attribute views ${X_1, ..., X_b}$, and we construct a $G_X$ for each of them. We've revised Section 3.1 accordingly to clarify the construction of $G_X$.
>
> **W2:** The presentation of the paper is confusing, I cannot see what is the final objective function.
>
> **Response:** To determine view weights for the aggregated multi-view Laplacian $L^*$, the optimization objectives are eigengap and connectivity derived from $L^*$ as in Eq. 3-4.
> * Eigengap: $max_w \frac{\lambda_{k+1}(L^*)}{\lambda_{k}(L^*)}$
> * Connectivity: $max_w \lambda_2(L^*)$
>     * $L^*=\sum_{i=1}^z w_iL_i$
>     * $\lambda_i(L^*)$ denotes the $i$-th smallest eigenvalue of $L^*$
>
> Equations 3-4 in our paper have been revised to emphasize the objective formulation.
>
> In the downstream tasks, spectral clustering minimizes the normalized cut as follows, where $C$ is a normalized cluster assignment matrix.
> * $NCut(C) = trace(C^TL^*C)$
>
> NetMF embedding algorithm performs factorization as follows, which is proved equivalent to optimizing the DeepWalk model. The matrix $M$ is approximated via eigen-decomposition of $L^*$.
> * $XX^T=trunclog(M),\ M=\frac{vol(G)}{bT}(\sum^T_{t=1}P^t)D^{-1}$

---

### Meta-Review · Area_Chair_NT2X · 2023-12-05

**Metareview:**

The paper studies the problem of clustering multi-view graphs. Multi-view graphs are common in practice and so designing good algorithms for them is important. The paper uses a spectral approach to introduce a new algorithm in this setting.

The problem studied in the paper is interesting and the paper contains some interesting insights although it is not ready for publication. In particular, during the reviewing discussion the following weaknesses were highlighted:

- the algorithm presented in the paper is not particularly novel

- the theoretical results presented are not too surprising or technically interesting

- the experimental results are a bit unconvincing

Overall, the paper is interesting but it is below the acceptance bar of ICLR.

**Justification For Why Not Higher Score:**

The paper has fundamental weaknesses and it is not novel enough

**Justification For Why Not Lower Score:**

N/A

---

### Decision · Program_Chairs · 2024-01-16

Reject